# Development and Validation of a Risk Scoring Tool for Bronchopulmonary Dysplasia in Preterm Infants Based on a Systematic Review and Meta-Analysis

**DOI:** 10.3390/healthcare11050778

**Published:** 2023-03-06

**Authors:** Zhumei Yu, Lili Wang, Yang Wang, Min Zhang, Yanqin Xu, Annuo Liu

**Affiliations:** 1Department of Neonatology, the First Affiliated Hospital of Anhui Medical University, Hefei 230022, China; 2School of Nursing, Anhui Medical University, Hefei 230032, China

**Keywords:** prematurity, low birth weight, bronchopulmonary dysplasia, meta-analysis, external validation, prediction, risk scoring tool

## Abstract

**Background:** Bronchopulmonary dysplasia (BPD) is the most common serious pulmonary morbidity in preterm infants with high disability and mortality rates. Early identification and treatment of BPD is critical. **Objective:** This study aimed to develop and validate a risk scoring tool for early identification of preterm infants that are at high-risk for developing BPD. **Methods:** The derivation cohort was derived from a systematic review and meta-analysis of risk factors for BPD. The statistically significant risk factors with their corresponding odds ratios were utilized to construct a logistic regression risk prediction model. By scoring the weights of each risk factor, a risk scoring tool was established and the risk stratification was divided. External verification was carried out by a validation cohort from China. **Results:** Approximately 83,034 preterm infants with gestational age < 32 weeks and/or birth weight < 1500 g were screened in this meta-analysis, and the cumulative incidence of BPD was about 30.37%. The nine predictors of this model were Chorioamnionitis, Gestational age, Birth weight, Sex, Small for gestational age, 5 min Apgar score, Delivery room intubation, and Surfactant and Respiratory distress syndrome. Based on the weight of each risk factor, we translated it into a simple clinical scoring tool with a total score ranging from 0 to 64. External validation showed that the tool had good discrimination, the area under the curve was 0.907, and that the Hosmer–Lemeshow test showed a good fit (*p* = 0.3572). In addition, the results of the calibration curve and decision curve analysis suggested that the tool showed significant conformity and net benefit. When the optimal cut-off value was 25.5, the sensitivity and specificity were 0.897 and 0.873, respectively. The resulting risk scoring tool classified the population of preterm infants into low-risk, low-intermediate, high-intermediate, and high-risk groups. This BPD risk scoring tool is suitable for preterm infants with gestational age < 32 weeks and/or birth weight < 1500 g. **Conclusions:** An effective risk prediction scoring tool based on a systematic review and meta-analysis was developed and validated. This simple tool may play an important role in establishing a screening strategy for BPD in preterm infants and potentially guide early intervention.

## 1. Introduction

Bronchopulmonary dysplasia (BPD), also known as chronic lung disease (CLD), is one of the most common complications in preterm infants [1]. Since 1990s, rapid advances in perinatal and neonatal care technologies have significantly improved the birth rate and survival rate of extremely preterm infants (EPIs) and very low birth weight infants (VLBWIs). Unfortunately, the incidence of BPD have increased [2,3]. Children with BPD have a high rate of re-hospitalization [4]. Severe BPD is associated with an increased risk of death, and survivors suffer from short and long-term sequelae, including respiratory morbidity and neurodevelopmental impairment [5,6,7].

BPD is an abnormal development of the lungs and pulmonary vasculature caused by multiple prenatal and postnatal risk factors. Although several individual risk factors have been identified, the interaction between these factors and their relationship are controversial [8]. Moreover, since there is no safe and effective treatment for BPD [9,10], it remains one of the most common and intractable diseases in the neonatal intensive care unit (NICU). In the early stages of life, predicting the greatest risk of BPD helps develop targeted prevention strategies before BPD is diagnosed in preterm infants, which is critical to reducing the incidence of BPD and improving prognosis.

In recent years, many scholars have devoted themselves to exploring and establishing suitable risk prediction models for BPD [11]. Most of the existing models were based on single-center observational research and analysis, which faced problems such as small sample size, unconvincing influencing factors, and availability of some predictors [12,13,14]. In addition, some BPD prediction models use relatively complex mathematical model calculations, and few models have undergone independent external validation or calibration [15,16,17,18]. The American Institute of Child Health and Human Development and the Neonatal Research Network (NICHD-NRN) published a web-based BPD prediction estimator [19]. However, this estimator lacked validation studies of Asian ethnicity. What is more, the population used by this estimator was limited to less than 1250 g and 23 to 30 weeks’ gestation, respectively, which accounted for only a small proportion of the preterm infant population [13,20]. Therefore, there is still a need to update predictive models in this area of research. This study aimed to develop and externally validate a simple early risk prediction scoring tool for BPD in preterm infants based on a systematic review and meta-analysis, and to provide an objective and effective screening tool for BPD in preterm infants.

## 2. Research Design and Methods

### 2.1. Meta-Analysis

#### 2.1.1. Design

This study was strictly performed in accordance with the Meta-analysis of Observational Studies in Epidemiology (MOOSE) guidelines [21] and the Preferred Reporting Items for Systematic Reviews and Meta-analyses (PRISMA) guidelines [22].

#### 2.1.2. Literature Search Strategy

PubMed, Web of Science, Embase and Cochrane Library were comprehensively searched to obtain relevant original English literature on risk factors for BPD from January 1990 to May 2022. Search strategies used combinations of the following search terms: (Infant, Premature [MeSH] OR Infant, preterm [tw]) OR (Infant, low birth weight [MeSH]) AND (bronchopulmonary dysplasia [MeSH] OR bpd [tw] OR chronic lung disease [tw]) AND (Risk Factors [MeSH] OR predictive factors [tw] OR dangerous factors [tw]). All relevant free search terms were searched for potential articles and used the snowball method to track references included in this meta-analysis. Two researchers (Z.Y. and M.Z.) screened the literature independently in strict accordance with the inclusion and exclusion criteria. Any differences in the search process were fully discussed with the third researcher (A.L.) until consensus was reached.

#### 2.1.3. Inclusion and Exclusion Criteria

Inclusion criteria: (1) preterm infants with gestational age (GA) < 32 weeks and/or birth weight (BW) < 1500 g; (2) studies were case-control or cohort studies based on original data; (3) the odds ratio (OR) and 95% confidence interval (95% CI) of the risk factor could be obtained directly or through data transformation; (4) the outcome was BPD; (5) articles published in English; (6) the study was published in a peer-reviewed journal.

Exclusion criteria: (1) the original study was not available; (2) the definition of BPD was unclear; (3) the outcomes were BPD or death, and data from separate categories of BPD (mild, moderate, and severe); (4) studies limited to gene polymorphisms; (5) the research types were animal studies, reviews, commentaries, editorials, case reports or republished articles.

#### 2.1.4. Data Extraction and Quality Assessment

Data extraction: Two researchers (Y.X. and Z.Y.) independently extracted data from relevant studies using a pre-determined data extraction form, and entered these into an Excel spreadsheet after a third researcher (Y.W.) checked data extraction for accuracy and completeness. The following information was extracted: (1) basic information includes first author, published year, country, number of centers, research time, study design, and (2) baseline characteristics include BW and GA, the number of case groups and control groups, and definition of BPD.

Quality assessment: The Newcastle Ottawa Scale (NOS) [23] was used to evaluate the quality of the included literature, which was divided into three parts: selectivity (4 points), comparability (2 points) and outcome or exposure (3 points), and the higher total score (≥7) indicated a higher quality of literature. Two researchers (M.Z. and Z.Y.) independently completed the quality evaluation and cross-checked the results. Any differences were resolved through a full discussion with the third researcher (L.W.).

#### 2.1.5. Statistical Analysis

The statistical analysis of the data was conducted using the statistical software “STATA17.0”. In order to ensure the stability of meta-analysis results, the risk factors were not analyzed if less than 3 articles were included. OR with corresponding 95% CI was calculated to estimate the statistical outcomes. Statistical heterogeneity was assessed and quantified using the Cochran Q statistic and I^2^ statistic. When the heterogeneity test indicated no significant (*p* > 0.1, I^2^ < 50%), a fixed-effects model was applied; otherwise, a random-effects model was used. Sensitivity analysis and subgroup analysis were performed to test the stability of the results and explore the source of heterogeneity, respectively. Begg’s test and Egger’s test were used to investigate publication bias. A 2-tailed *p* < 0.05 was considered statistically significant.

### 2.2. Development of the Risk Scoring Tool

We developed a prediction scoring tool based on a systematic review and meta-analysis. The appropriate OR with 95% CI of each risk factor in the model was used to calculate the corresponding *β*-coefficient, and the formula β=ln(OR). The constant term α is the natural logarithm of the ratio of the incidence and non-incidence of BPD in a certain period, i.e., α=ln(p/(1-p)). The logarithmic function of the BPD prediction model is $logit(P)=\alpha +\beta_{1}X_{1}+\beta_{2}X_{2}+\beta_{3}X_{3}+\cdots \cdots +\beta_{n}X_{n}$, $\beta_{n}$ represents the regression coefficient for the *n*_th_ risk factor, and $X_{n}$ represents the *n*_th_ risk factor. To develop a simple risk scoring tool, we calculated the corresponding score by dividing each regression coefficient $\beta_{n}$ by the minimum $\beta_{min}$ of the absolute value, and rounding it to the nearest whole number. The scores for each risk factor were added to calculate the total score. As the cumulative score increases, the risk of developing BPD increases.

### 2.3. Validation of the Risk Scoring Tool

#### 2.3.1. Validation Cohort

The validation cohort was from a population retrospective cohort of preterm infants hospitalized in NICU, the First Affiliated Hospital of Anhui Medical University, between 1 June 2017 and 1 June 2022. Preterm infants included in this cohort must meet the following criteria: (1) GA at birth < 32 weeks and/or BW < 1500 g; (2) admission to NICU within 24 h after birth; and (3) survival time ≥ 28 days. The following preterm infants were excluded: (1) complicated congenital heart disease, genetic metabolic disease, diaphragmatic hernia or congenital respiratory malformation; (2) missing or incomplete data; or (3) death before fulfilling the BPD definition. The study protocol was approved by the Committee of the First Affiliated Hospital of Anhui Medical University (No. Quick-PJ2022-08-31). Due to the retrospective nature of this study, informed consent was waived.

#### 2.3.2. Definitions

The definition criteria of some clinical indicators in the validation cohort were shown as follows:(1)The diagnosis of BPD was acquired according to the 2018 definition proposed by NICHD [24], preterm infants < 32 weeks’ gestation with radiographically confirmed persistent parenchymal lung disease, and at 36 weeks postmenstrual age (PMA) requiring a certain respiratory support and fraction of inspired oxygen (FiO_2_) for ≥3 consecutive days to maintain arterial oxygen saturation in the 90–95% range.(2)Maternal hypertensive disorders (MHD) included chronic hypertension (with blood pressure values ≥ 140/90 mmHg of any cause, hypertension was documented before pregnancy or before 20 weeks’ gestation), gestational hypertension (systolic blood pressure ≥ 140 mmHg and/or diastolic blood pressure ≥ 90 mmHg occurring after 20 weeks’ gestation), preeclampsia (systolic blood pressure ≥ 140 mmHg and/or diastolic blood pressure ≥ 90 mmHg along with proteinuria higher than 300 mg in 24 h, detected after 20 weeks’ gestation in a previously normotensive woman) and eclampsia [25].(3)Chorioamnionitis (CA) was defined as clinical CA with or without histological proven CA [26].(4)The administration of antenatal steroids was defined as a single course of 4 doses of 6 mg of dexamethasone given intramuscularly 12 h apart.(5)Small for gestational age (SGA) was defined as a BW below the 10th percentile or 2 standard deviations below the mean weight for the same GA [27].(6)Respiratory distress syndrome (RDS) was defined as a clinical symptom (progressive dyspnea several hours after birth) consistent with chest X-ray findings [28].(7)Surfactant treatment was given as early as possible according to the neonatal respiratory condition after birth combined with evidence of RDS [29].(8)Sepsis was defined as an infection that can be proven by the culture of pathogenic bacteria (including bacteria and fungi) in blood, cerebrospinal fluid, or other sterile cavities under the premise of abnormal clinical signs and symptoms [30]. Based on the time of onset, the onset time was split into early-onset sepsis ≤ 3 days, and late-onset sepsis > 3 days.

#### 2.3.3. Statistical Analysis

The receiver operating characteristic (ROC) curve and area under the ROC curve (AUC) were used to evaluate the prediction power of the model. The value of AUC ranges from 0.5 to 1.0, and the greater the value, the better the prediction accuracy. The calibration curve was drawn by bootstrapping with 1000 resamples. The Hosmer−Lemeshow test was used to evaluate the goodness-of-fit of the model, and *p* > 0.05 was considered to show good model consistency. The classification of the model was evaluated using a decision curve analysis (DCA), reflecting the net benefit of the model to patients.

Youden’s J statistic (Youden’s index = sensitivity + specificity − 1) was used to calculate the cut-off points, and *p* < 0.05 was considered statistically significant. Patients were divided into low, low-intermediate, high-intermediate, and high groups based on the obtained frequencies of BPD using different risk scores. Statistical analyses were performed with SPSS 26.0 software and R version 4.1.2 software.

## 3. Results

### 3.1. Meta-Analysis

#### 3.1.1. Literature Search Results

A total of 12,915 potentially relevant studies were initially identified. By eliminating duplicate articles and screening titles and abstracts, 58 studies [2,12,13,15,16,17,20,25,31,32,33,34,35,36,37,38,39,40,41,42,43,44,45,46,47,48,49,50,51,52,53,54,55,56,57,58,59,60,61,62,63,64,65,66,67,68,69,70,71,72,73,74,75,76,77,78,79,80] met the inclusion and exclusion criteria after full-text examination (Appendix A). The flow diagram of the study selection process is shown in Figure 1.

#### 3.1.2. Study Characteristics

A total of 58 studies [2,12,13,15,16,17,20,25,31,32,33,34,35,36,37,38,39,40,41,42,43,44,45,46,47,48,49,50,51,52,53,54,55,56,57,58,59,60,61,62,63,64,65,66,67,68,69,70,71,72,73,74,75,76,77,78,79,80] were included in this review, including 26 multicenter studies [2,17,20,25,31,32,34,35,37,40,41,42,43,48,50,51,53,54,55,56,58,66,69,70,75,77], 32 single-center studies [12,13,15,16,33,36,38,39,44,45,46,47,49,52,57,59,60,61,62,63,64,65,67,68,71,72,73,74,76,78,79,80], 49 cohort studies [2,12,13,17,20,25,31,32,33,34,35,36,37,39,40,41,42,43,44,45,46,48,49,50,51,53,54,56,57,58,59,60,61,62,63,64,65,66,67,69,70,71,72,73,74,75,76,77,80] and nine case-control studies [15,16,38,47,52,55,68,78,79]. The studies were published between 1998 and 2022, and the research period was from 1990 to 2020. A total of approximately 83,034 preterm infants were screened, including 25,221 BPD cases. The characteristics of the studies are shown in Appendix A. The NOS scale was used to assess the quality of 58 included studies, and the scores of each study were ≥7 points, indicating a higher overall quality (Appendix A).

#### 3.1.3. Results of the Meta-Analysis

After listing the risk factors for BPD presented in the 58 studies [2,12,13,15,16,17,20,25,31,32,33,34,35,36,37,38,39,40,41,42,43,44,45,46,47,48,49,50,51,52,53,54,55,56,57,58,59,60,61,62,63,64,65,66,67,68,69,70,71,72,73,74,75,76,77,78,79,80], each factor was defined and categorized. The selected risk factors were divided into three categories: prenatal factors, intrapartum factors, and postpartum factors. 24 different risk factors were selected for analysis based on the risk factors that were investigated in at least three articles (Appendix A). The meta-analysis showed that the pooled effects of 19 risk factors were significant, as follows: CA, GA, BW, Sex, SGA, 5 min Apgar score, Delivery room intubation (DRI), Neonatal asphyxia, Mechanical ventilation (MV), Days of MV, MV > 7 days, Postnatal steroids, Surfactant, Patent ductus arteriosus (PDA), RDS, Sepsis, Intraventricular hemorrhage (IVH), Necrotizing enterocolitis (NEC), Pulmonary air leak. Details of each risk factor are presented in Appendix A.

#### 3.1.4. Publication Bias

The 12 risk factors assessed by the Begg’s test and Egger’s test suggested publication or selection bias. The results were: CA, GA, Sex, DRI, MV, Days of MV, Surfactant, PDA, RDS, Sepsis, NEC, Pulmonary air leak. Details of publication bias for all risk factors are presented in Appendix A.

### 3.2. Study Populations

#### 3.2.1. Derivation Cohort

The derivation cohort was derived from 58 studies [2,12,13,15,16,17,20,25,31,32,33,34,35,36,37,38,39,40,41,42,43,44,45,46,47,48,49,50,51,52,53,54,55,56,57,58,59,60,61,62,63,64,65,66,67,68,69,70,71,72,73,74,75,76,77,78,79,80] included in a systematic review and meta-analysis. Study populations were from North America [34,36,60,65,69,70,71] (the United States, Canada), South American [35,43,45,55,63] (Colombia, Argentina, Chile, Peru, Uruguay, Paraguay, Brazil), Europe [13,17,25,31,33,37,38,40,48,49,50,51,53,64,76,80] (the United Kingdom, Germany, Finland, Switzerland, Italy, Slovenia, Portugal, Spain, Belgium, Denmark, France, Netherlands, Poland, Sweden.), Asia [2,12,15,16,20,39,41,44,46,47,52,56,57,58,59,62,66,67,72,73,74,75,77,78,79] (China, China Taiwan, South Korea, Japan, Malaysia, Israel, Turkey, Saudi Arabia, Iran) and Oceania [32,42,54,61,68] (Australia, New Zealand). The derivation cohort included approximately 83,034 preterm infants, including 41,179 (49.59%) males. GA was roughly distributed between 22 and 37 weeks and BW was roughly distributed between 280 and 2445 g in all patients. There were approximately 25,221 cases of BPD, and the cumulative incidence of BPD was 30.37%. Detailed basic characteristics of preterm infants in the derivation cohort are shown in Appendix A.

#### 3.2.2. Validation Cohort

A total of 876 preterm infants with GA < 32 weeks and/or BW < 1500 g were preliminarily screened. Preterm infants with congenital anomalies potentially causing oxygen dependency (*n* = 21), admission exceeding 24 h after birth (*n* = 5), and death before fulfilling the BPD definition (*n* = 83) were excluded. Finally, 767 preterm infants were included in the validation cohort (Figure 2).

The validation cohort included 767 preterm infants, of whom 399 (52.0%) were males. In total, 49.9% of the mothers were treated with antenatal steroids, 15.5% had CA, 43.5% of infants were treated with surfactants after delivery, the mean GA was 30.14 ± 1.79 weeks, the mean BW was 1306.04 ± 314.28 g, and the mean 5 min Apgar score was 8.63 ± 1.89. There were 185 cases of BPD, and the cumulative incidence of BPD was 24.12%. Detailed basic characteristics of preterm infants in the validation cohort are shown in Appendix A.

### 3.3. Risk Scoring Tool Development

A total of 24 risk factors were identified through a systematic review and meta-analysis. Due to *p* ≥ 0.05, the following five risk factors were excluded: MHD, Antenatal steroids, PROM, Caesarean section and ROP. Due to heterogeneity, partial results were selected from appropriate subgroup analyses or sensitivity analyses. To identify early preterm infants that are at high-risk of developing BPD, the following nine risk factors were included: CA, GA, BW, Sex, SGA, 5 min Apgar score, DRI, Surfactant, RDS. Forest plots are presented in Appendix A. Sensitivity and/or subgroup analyses are presented in Appendix A. Finally, nine risk factors were included in this prediction model (Table 1), as CA (OR = 3.56, 95% CI [2.49, 5.11], *p* = 0.000); GA per 1 week increase (OR = 0.64, 95% CI [0.62, 0.67], *p* = 0.000); BW per 100 g increase (OR = 0.78, 95% CI [0.76, 0.80], *p* = 0.000); Sex (OR = 1.46, 95% CI [1.39, 1.54], *p* = 0.000); SGA (OR = 4.78, 95% CI [3.88, 5.88], *p* = 0.000); 5 min Apgar score per 1 point increase (OR = 0.71, 95% CI [0.64, 0.78], *p* = 0.000); DRI (OR = 2.77, 95% CI [2.27, 3.39], *p* = 0.000); Surfactant (OR = 3.59, 95% CI [2.90,4.45], *p* = 0.000); RDS (OR = 5.08, 95% CI [4.06, 6.35], *p* = 0.000). The risk score was calculated as follows:(1)$$\begin{array}{l} Logit(P)=-1.15+1.270\times CA-0.446\times GA-0.248\times BW+0.378\times S\mathrm{ex}+1.564\times SGA\\ \;\;\;\;-0.342\times 5\;\min\;Apgar\;score+1.019\times DRI+1.278\times Surfactant+1.625\times RDS \end{array}$$

We translated this into a simple risk prediction scoring tool with an overall score ranging from 0 to 64 points. The established early prediction scoring tool for BPD in preterm infants was classified in detail as follows: CA (no = 0, yes = 5); GA (≥32 = 0, 31–31^+6^ = 2, 30–30^+6^ = 4, 29–29^+6^ = 6, 28–28^+6^ = 8, 27–27^+6^ = 10, 26–26^+6^ = 12, 25–25^+6^ = 14, 24–24^+6^ = 16, <24 = 18); BW (≥1500 = 0, 1400–1499 = 1, 1300–1399 = 2, 1200–1299 = 3, 1100–1199 = 4, 1000–1099 = 5, 900–999 = 6, 800–899 = 7, 700–799 = 8, 600–699 = 9, 500–599 = 10, <500 = 11); Sex (female = 0, male = 2); SGA (no = 0, yes = 6); 5 min Apgar score (≥8 = 0, 7 = 1, 6 = 2, 5 = 3, 4 = 4, 3 = 5, ≤2 = 6); DRI (no = 0, yes = 4); Surfactant (no = 0, yes = 5); RDS (no = 0, yes = 7) (Appendix A).

### 3.4. Risk Scoring Tool Validation

The ROC curves were drawn for the validation cohort. The AUC value was (0.907, 95% CI [0.883, 0.931]) predicted by combined indicator (CA, GA, BW, Sex, SGA, 5 min Apgar score, DRI, Surfactant, RDS), sensitivity was 0.897, and specificity was 0.873 (Table 2). The ROC curves showed that the prediction performance of combined indicators was better than that of single indicator (Figure 3). Meanwhile, The Hosmer–Lemeshow test demonstrated that the tool showed a good fit (*p* = 0.3572), and the calibration curve showed excellent agreement between the predicted and actual observations, indicating that the tool had good consistency (Figure 4). Furthermore, the DCA showed that the red curve is higher than the two extreme lines, reflecting a significant net benefit of the tool (Figure 5).

Based on the sensitivity of 0.897 and the specificity of 0.873, we concluded that the Youden index maximum was 0.770 and the corresponding optimal cut-off value was 25.5, indicating that patients with risk score ≥ 25.5 points were most likely to develop BPD (Appendix A). Based on the obtained frequencies of BPD using different risk scores, we further distinguished four risk groups with different outcomes: low (risk score ≤ 12), low-intermediate (13 ≤ risk score ≤ 25), high-intermediate (26 ≤ risk score ≤ 44), and high (risk score ≥ 45). The validation cohort ware used to further verify the prevalence of BPD and evaluate its prediction effectiveness. The results showed that the actual prevalence rates of each risk group were 0.5% in the low-risk group, 5.5% in the low-intermediate group, 67.0% in the high-intermediate group, and 94.7% in the high-risk group. Compared with the low-risk group, the OR values for developing BPD in the low-intermediate, high-intermediate and high-risk groups were (11.592, 95% CI [1.535, 87.520], *p* = 0.000), (403.452, 95% CI [55.439, 2936.093], *p* = 0.000) and (3582.000, 95% CI [214.906, 59703.777], *p* = 0.000), respectively (Table 3). The final established risk prediction scoring tool for BPD is detailed in Table 4. This simple tool is suitable for preterm infants with GA < 32 weeks and/or BW < 1500 g.

## 4. Discussion

Despite significant advances in neonatology over the past several decades, the incidence of BPD has actually been rising worldwide [3]. Since BPD is a heterogeneous disease that makes medical management difficult, it is important to prevent and predict the disease early, rather than waiting for treatment and care after irreversible changes have occurred. In this study, we integrated 58 high-quality case-control studies and cohort studies for systematic review and meta-analysis, and identified independent risk factors for developing BPD in premature infants. In order to predict the disease early in the onset (within the first week of life), we selected some risk factors and finally established a BPD risk-scoring tool for preterm infants. The nine predictors of the scoring tool were CA, GA, BW, Sex, SGA, 5 min Apgar score, DRI, Surfactant and RDS. Based on the total score of each predictor, the tool can be used to early assess the risk of developing BPD in preterm infants.

Previous data were contradictory, and the relationship between CA and BPD showed either an increase, decrease, or no effect [81,82,83,84]. Our study showed that, in preterm infants, exposure to CA is associated with a higher risk of developing BPD. Intrauterine infection and/or inflammation disturbed fetal lung development, and leads to the initiation and progression of BPD [10,85]. Therefore, strengthening perinatal care and increasing the proportion of antenatal care for pregnant women may reduce the incidence of BPD by reducing the chance of maternal infection.

The results of this study showed that GA and BW were inversely proportional to the incidence of BPD, which is consistent with the results of many studies [1,56,86]. Lung development in the canalicular stage or saccular stage is vulnerable to oxidative stress damage, and lung immaturity at birth is a key factor in the development of BPD. Nearly 80% of infants with BPD had a GA of 22–24 weeks and 95% were VLBWs [87]. Therefore, strengthening perinatal care management and actively preventing and reducing preterm birth are essential to prevent BPD. Rocha et al. [88] reported that SGA was an independent risk factor for BPD, which is consistent with the results of this study. The effect of SGA on the development of BPD also seems to be related to the inhibition of lung growth, possibly involving changes in angiogenesis. A Sheep model confirmed reduced alveolarization and abnormal pulmonary vascular development in fetal growth restriction fetuses [89]. In the current study, males were found to have a higher risk of developing BPD [90]. There are sex-dependent differences in lung development in extremely preterm infants [91], and different genders also showed different gene expression patterns in the BPD mouse model [92]. The Apgar score is one of the oldest and most acceptable methods for assessing neonatal status after birth in the delivery room. A low Apgar score of 5 min or more is often an indicative criterion for asphyxia [93]. This is not only closely associated with neonatal mortality and organ dysfunction [94], but also significantly increases the incidence of BPD [15]. DRI was often used as a treatment for resuscitation in the delivery room, which may have potential harmful effects on preterm infants, including early barovolu-trauma, early exposure to high partial pressures of ambient oxygen, and increased risk of bacterial colonization of the alveoli and lower airways [95]. On the other hand, DRI may indicate a greater likelihood of subsequent mechanical ventilation in preterm infants. Multiple studies have shown that centers with high rates of DRI tend to have higher rates of ventilation and BPD [51,96]. These findings suggest the need to strengthen the training in asphyxia resuscitation strategy for obstetric and neonatal health care workers, and reduce the incidence of DRI by improving the quality of perinatal management, thereby reducing adverse outcomes of preterm infants [97].

The surfactant was a complex mixture of phospholipids and proteins, the synthesis of which depends on the differentiation of alveolar type 2 epithelial cells (AT2), which usually occurs in the third trimester. Therefore, incomplete differentiation of AT2 cells in preterm infants results in insufficient pulmonary surfactant, which contributes to the pathogenesis of RDS [98]. For years, invasive mechanical ventilation (IMV) initiation of infants in the first few hours of life was the primary treatment of VLBWIs with RDS [99]. However, improper use of IMV may lead to ventilator-related lung injury (VILI) [100], thereby increasing the incidence of BPD [101]. Although the introduction of exogenous surfactants has significantly reduced mortality of EPIs with RDS, it did not reduce the overall proportion of preterm infants with BPD. Therefore, lung protective ventilation strategies need to be emphasized to reduce unnecessary lung injury. The European consensus guidelines recommend that early routine use of continuous positive airway pressure, combined with selective use of surfactant, is the best treatment for most preterm infants with early or mild RDS [29].

At present, a number prediction models for BPD in preterm infants have been proposed. However, almost all existing models were based on observational studies, and there are no evidence-based studies. Most were developed in specific centers and populations, while lacking independent external validation. The predictive models tended to perform well in modeled data, but their accuracy was significantly reduced when other external validations were performed, so generalizability was limited [11,18]. The current online NICHD-NRN estimator is not suitable for all races, and the GA and BW limits for the applicable population are severe. We searched relevant studies in various countries, avoiding the problem of lack of representation of single-center research samples. At the same time, strict inclusion and exclusion criteria were adopted, and the quality of the final included literature was assessed to ensure reliability. As for the results of meta-analysis, we selected the risk factors supported by ≥ 3 studies and conducted sensitivity analysis and bias assessment at the same time to ensure the stability and credibility of the results. Based on the parameters obtained from meta-analysis, the risk-scoring tool of BPD was finally established. In additon, tools that were difficult to use or explain have little value in a busy NICU. This scoring tool can be used at an early stage, indicators are easily available, and calculations are simple. Furthermore, we selected an independent unit database for retrospective cohort analysis and validation, so as to ensure the scientific accuracy of the tool. External validation confirmed that the risk scoring tool had an AUC of 0.906, when the best cut-off value was 25.5, and its sensitivity and specificity were 0.897 and 0.873, respectively. Moreover, the results of the Hosmer-Lemeshow test, calibration curve and DCA all indicated that our tool had significant predictive ability and clinical value. Another advantage of this study is risk stratification of at-risk populations. This not only allows clinicians to quantify the risk of BPD in preterm infants to benefit counseling of families, but also helps investigators optimize clinical trials, target personalized management in high-risk preterm infants and save medical resources.

The present study has certain limitations. First, the 58 articles included came from different regions and races, which inevitably displayed some heterogeneity. Although we conducted subgroup analysis and sensitivity analysis, the sources of heterogeneity of some risk factors could not be determined. Second, every risk factor included ≥ 3 studies, and although the stability of the results was guaranteed to a certain extent, other clinical indicators cannot be included in the analysis. Third, due to the different risk factors for different outcomes in the meta-analysis, we excluded the outcomes BPD or death, and severity of BPD (mild, moderate, and severe). Finally, the derivation cohort of this study came from populations from different countries, while the validation cohort was only from preterm infants from one hospital in China, which may be an issue in applying data generated from mainly Caucasian populations. However, ethnic and genetic differences in the prediction of BPD cannot be ignored, the large data set may provide an important opportunity to evaluate outcome differences in Caucasian vs. Asian cohorts. In the future, we hope to continue to update our model, including adding outcomes that predict the severity of BPD, and conduct multi-center external validation to further provide a scientific and objective reference tool for early clinical identification and prevention of BPD.

## 5. Conclusions

We have provided and validated an easy-to-use, low-cost, early predicting-capabilities and accurate BPD risk-scoring tool based on a systematic review and meta-analysis to identify and stratify high-risk preterm infants for BPD. This tool may play an important role in clinical practice and generating targeted screening strategies for BPD in preterm infants.

## Figures and Tables

**Figure 1 healthcare-11-00778-f001:**
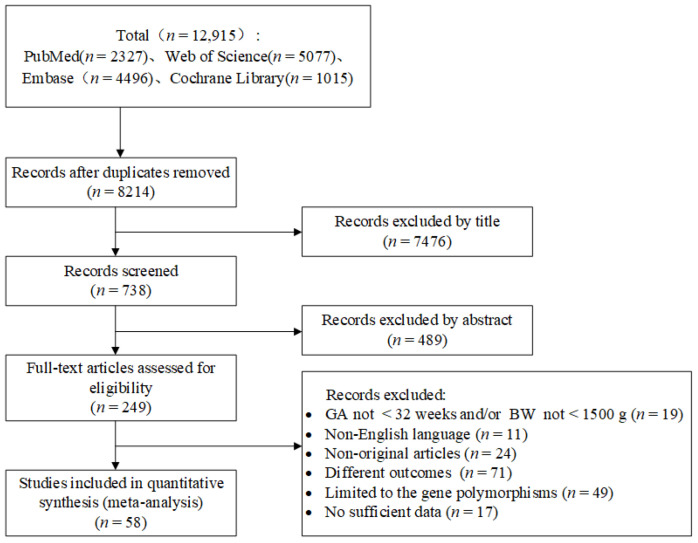
Flow diagram of study screening. GA: gestational age; BW: birth weight.

**Figure 2 healthcare-11-00778-f002:**
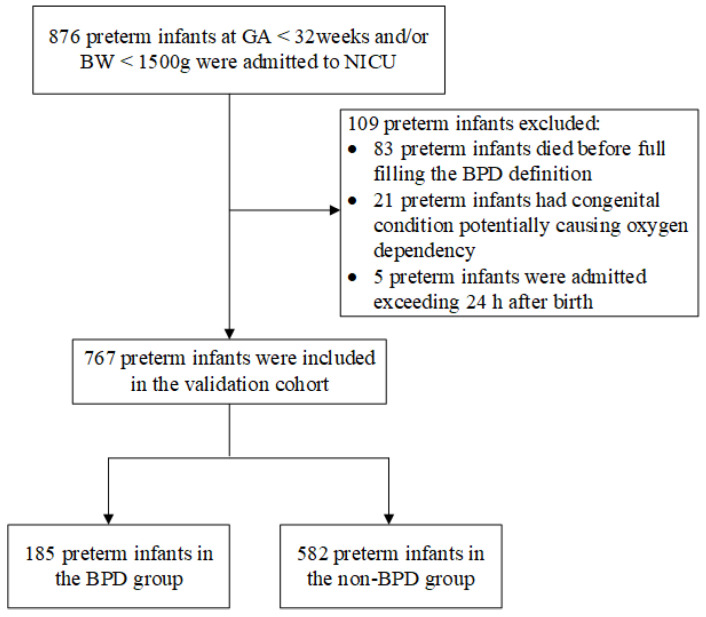
Flow diagram showing process for selection of preterm infants in the validation cohort between 1 June 2017 and 1 June 2022. GA: gestational age; BW: birth weight; NICU, neonatal intensive care unit; BPD, bronchopulmonary dysplasia.

**Figure 3 healthcare-11-00778-f003:**
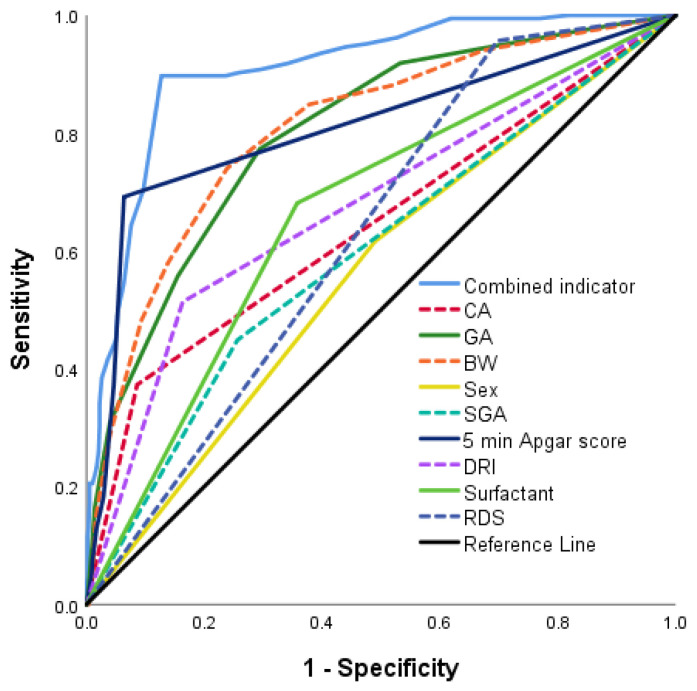
Receiver operating characteristic curve of BPD prediction (area under the curve = 0.907). Abbreviations: BPD, bronchopulmonary dysplasia; CA, Chorioamnionitis; GA, gestational age; BW, birth weight; SGA, small for gestational age; DRI, delivery room intubation; RDS, respiratory distress syndrome.

**Figure 4 healthcare-11-00778-f004:**
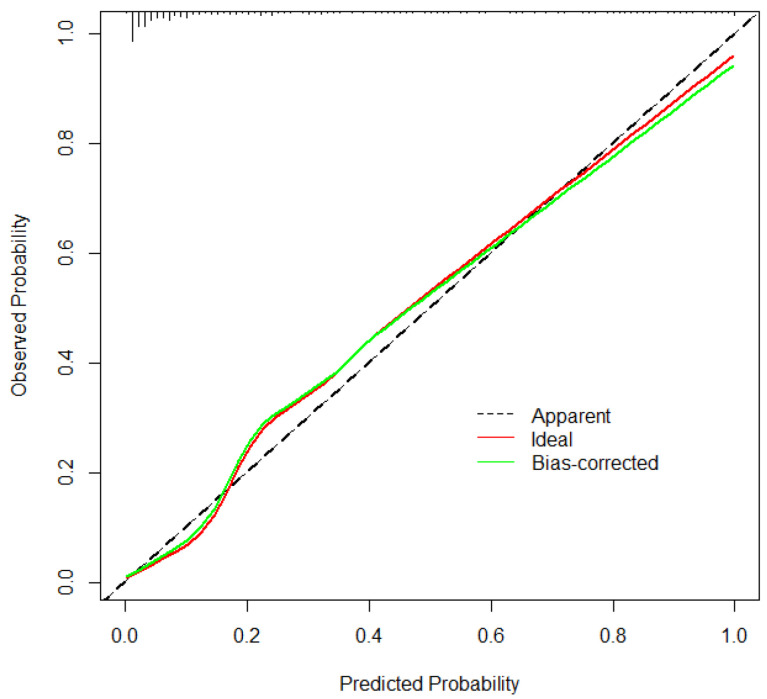
Calibration curve for predicting probability of BPD in preterm infants. Abbreviations: BPD, bronchopulmonary dysplasia.

**Figure 5 healthcare-11-00778-f005:**
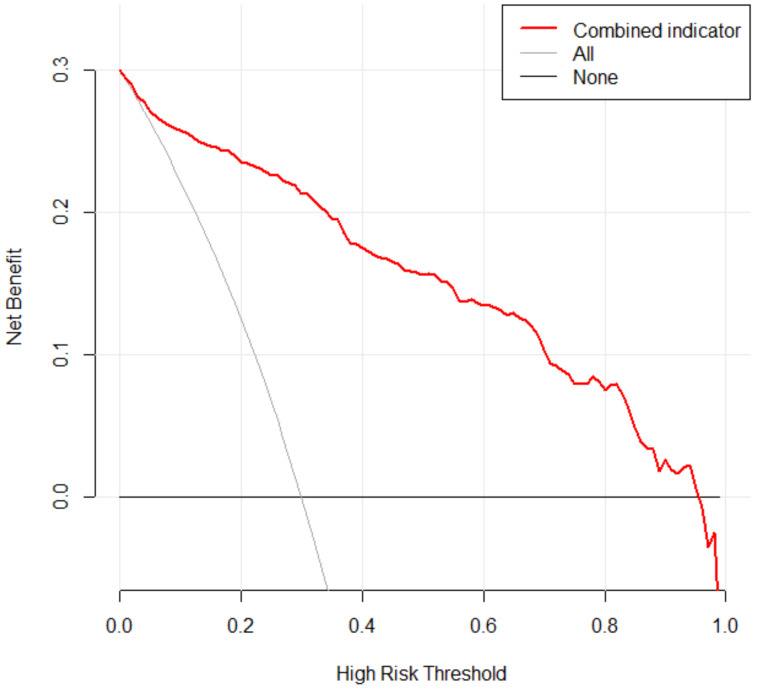
Decision curve analysis in prediction of BPD in preterm infants. Abbreviations: BPD, bronchopulmonary dysplasia.

**Table 1 healthcare-11-00778-t001:** Risk stratification, OR (95% CI), β-coefficient, and scores of risk factors included in the BPD risk prediction model.

Risk Factors	Risk Stratification	OR	95% CI	β-Coefficient	Scores
CA	yes/no	3.56	[2.49, 5.11]	1.270	5
GA	per 1 week increase	0.64	[0.62, 0.67]	−0.446	−2
BW	per 100 g increase	0.78	[0.76, 0.80]	−0.248	−1
Sex	male/female	1.46	[1.39, 1.54]	0.378	2
SGA	yes/no	4.78	[3.88, 5.88]	1.564	6
5 min Apgar score	per 1 point increase	0.71	[0.64, 0.78]	−0.342	−1
DRI	yes/no	2.77	[2.27, 3.39]	1.019	4
Surfactant	yes/no	3.59	[2.90, 4.45]	1.278	5
RDS	yes/no	5.08	[4.06, 6.35]	1.625	7

Abbreviations: OR, Odds ratio; CI, confidence interval; BPD, bronchopulmonary dysplasia; CA, Chorioamnionitis; GA, gestational age; BW, birth weight; SGA, small for gestational age; DRI, delivery room intubation; RDS, respiratory distress syndrome.

**Table 2 healthcare-11-00778-t002:** The AUC values predicted by each factor and combined indicator in the validation cohort.

Measurement Indicators	Sensitivity	Specificity	Youden Index	AUC	95% CI	*p*
Combined indicator	0.897	0.873	0.770	0.907	[0.883, 0.931]	0.000
CA	0.373	0.914	0.287	0.644	[0.594, 0.693]	0.000
GA	0.773	0.706	0.479	0.800	[0.764, 0.837]	0.000
BW	0.741	0.761	0.502	0.807	[0.770, 0.844]	0.000
Sex	0.616	0.510	0.126	0.563	[0.516, 0.610]	0.009
SGA	0.449	0.744	0.193	0.596	[0.548, 0.645]	0.000
5 min Apgar Score	0.692	0.936	0.628	0.809	[0.767, 0.851]	0.000
DRI	0.514	0.837	0.351	0.675	[0.627, 0.723]	0.000
Surfactant	0.681	0.643	0.324	0.662	[0.617, 0.707]	0.000
RDS	0.957	0.302	0.259	0.630	[0.588, 0.671]	0.000

Abbreviations: AUC, area under the curve; CI, confidence interval; CA, Chorioamnionitis; GA, gestational age; BW, birth weight; SGA, small for gestational age; DRI, delivery room intubation; RDS, respiratory distress syndrome.

**Table 3 healthcare-11-00778-t003:** Four risk groups stratified by risk score in the validation cohort.

Risk Stratification	Total (n = 767)	BPD (n = 185)	Prevalence Rate (%)	χ2	*p*	OR (95% CI)
Low	200	1	0.5			
Low-intermediate	327	18	5.5	8.944	0.000	11.592 [1.535, 87.520]
High-intermediate	221	148	67.0	202.852	0.000	403.452 [55.439, 2936.093]
High	19	18	94.7	194.485	0.000	3582.000 [214.906,59703.777]

Abbreviations: BPD, bronchopulmonary dysplasia; OR, Odds ratio; CI, confidence interval.

**Table 4 healthcare-11-00778-t004:** Bronchopulmonary dysplasia risk prediction scoring tool for preterm infants.

Bronchopulmonary Dysplasia Risk-Scoring Tool for Preterm Infants	
**Variable Category and Risk Factors**	**Points**
**Chorioamnionitis**	5
**Gestational age (GA) (select one)**	
≥32 weeks	0
24–31^+6^ weeks	2 × [32 − GA]
<24 weeks	18
**Birth weight (BW) (select one)**	
≥1500 g	0
500–1499 g	1 × [15 − BW/100]
<500 g	11
**Male**	2
**Small for gestational age**	6
**5 min Apgar score (select one)**	
≥8	0
3–7	1 × (8 − 5 min Apgar score)
≤2	6
**Delivery room intubation**	4
**Surfactant**	5
**Respiratory distress syndrome**	7
“[ ]” means ceiling function;	
Scoring risk ranges: 0–12, low; 13–25, low-intermediate; 26–44, high-intermediate; ≥45, high.	

## Data Availability

Source data from the study may be achieved from authors upon request.

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
