# Peer review of "Development and Validation of a Risk Scoring Tool for Bronchopulmonary Dysplasia in Preterm Infants Based on a Systematic Review and Meta-Analysis"

_healthcare, 2023, doi:10.3390/healthcare11050778_

Round 1

Reviewer 1 Report

Thank you for giving me the opportunity to review the article entitled “Development and validation of a risk scoring tool for bronchopulmonary dysplasia in preterm infants based on a systematic review and meta-analysis”

Abstract:

-          The authors use abbreviations not previously described and are not clear with the explanation of the result obtained. P.e. “The 9 predictors of this model were CA, GA, BW, Gender, SGA, 5 min Apgar score, DRI, Surfactant and RDS. Based on the weight of each risk factor, we translated it into a simple clinical scoring system with a total score ranging from 0 to 64. External validation showed that the model had good discrimination with AUC = 0.907. When the optimal cut-off value was 25.5, the sensitivity and specificity were 0.897 and 0.873, respectively”

-          “This tool is suitable for preterm infants with GA < 32 weeks and/or BW < 1500g” what tool are you referring to?

Research design and methods

Design

PRISMA recommendations used are old, there is a new update (PRISMA 2021)

Literature search strategy

What was the search strategy used?

The section “Risk factors for BPD in preterm infants” as well as the following sections belong to results obtained

The structure of the meta-analysis as well as the objectives that the authors intend to describe are not valid.

I advise reviewing the work and reediting what the objectives of it are. The development of the meta-analysis is incorrect

Reviewer 2 Report

In the paper Development and validation of a risk scoring tool for bronchopulmonary dysplasia in preterm infants based on a systematic review and meta-analysis a scoring tool used to predict the risk of bronchopulmonary dysplasia has been developed based off of a meta-analysis of 58 studies.  The strengths of the study are that it is a meta analysis that includes a large number of patients from multiple institutions and ethnicities, it is easy to use and understand and can be used at the bedside to counsel families about the risk of developing BPD early in their clinical course.  The authors also validated their model using their single center data.  The weaknesses of the study include that all of these risk factors are well known and have been previously published; there are at least 58 previously published papers that deal with this topic.  Because of the large sample size they have shown that some factors previously reported are not statistically significant and probably do not need to be taken into account.  While a scoring tool can be useful to counsel families, non-of the risk factors that they report can be acted on in a given patient.  The GA, BW, and sex cannot be changed.  Further they use an aggregate outcome of all grades of BPD, where a scoring tool that predicted a more specific outcome would be more useful.  Telling parents that there child may be discharged on 0.5L of O2 is significantly different than sending a child home with a trach/vent and gtube requiring long term mechanical ventilation. 

Abstract: Don’t use abbreviations in the abstract, specifically when talking about the 9 predictors in the model. 

Introduction:

The introduction should be shortened and more focused.  Specifically the 1st and 2nd paragraph talk about the pathogenesis of BPD and some outcomes which are not relevant to this paper.  Keep it focused on the need for a model and why previous models are inadequate.

Methods:

Why exactly did you exclude the severity of BPD from the analysis?  Predicting the severity of BPD would be much more useful than predicting BPD in general.  Jensen’s definition from 2019 stratified BPD into three grades (1, 2, 3), with a strong correlation with outcomes.  Did you look at models that predicted different grades of BPD?  How did they perform?

Results/Discussion:

At what age would you recommend using this scoring tool? As soon as the patient has been diagnosed with RDS?  Given that every other risk factor occur before, or at birth this would allow you to stratify patients within the first week of life.

Discussion:

Last sentence of the first paragraph:  This isn’t true.  Given that all of the predictors, CA, GA, BW, Sex, SGA, 5 min Apgar score, DRI, Surfactant and RDS are evaluated retrospectively and that they are all things that are either unmodifiable (ie Sex), or things that occur despite best medical care (CA, BW, SGA, ect), they cannot be intervened on to prevent BPD using your scoring method.  There is no intervention that can be chosen or used to modify the risk of BPD for a given infant once they have been given a score of “High”.

While it is true that improved training of obstetric and neonatal health care workers can improve outcomes, based off your model it is impossible to say why patients were intubated in the delivery room and draw conclusions from it.  Were they intubated for some other factor and DRI is a surrogate marker for this unknown factor that predisposed to BPD, or did the act of intubating in the DR itself contribute to BPD.  It is an association, but does not provide insight as to causality. 

Last sentence in the second paragraph.  In appropriate for this discussion and goes beyond the scope of your paper. 

Last sentence 3rd paragraph.  This is why the tool is useful clinically.  It can be used to counsel families about the risks of developing BPD.  Unfortunately BPD is a heterogenous disease and while this tool is useful in predicting the development of any type of BPD (aggregate Jensen grade 1, 2, 3), it is less useful in that it does not allow the clinician to make predictions about specific morbidities such as the need for a tracheostomy or long term mechanical ventilation.  Further it does not allow the clinician to develop corresponding interventions to ameliorate the risk of BPD.  Most ongoing research is already focused on reducing all of the risk factors in your model (except Sex).  Your model might be useful to see if a reduction in any one risk factor by a given intervention could lead to a reduction in the rates of BPD.

Conclusion:

Last sentence: as previously stated this tool for a given infant cannot provide information that would lead to interventions that would reduce the risk of BPD. 

General Comment:

Paper needs to be edited for grammar

Reviewer 4 Report

Thank you for the opportunity to review this manuscript.  I have the following suggestions to be considered to improve upon for readability and clarity purposes:

1.  The search terms for the review are said to be used in various combinations, etc.  Unfortunately, this description of the search method prevents readers from recreating the search on their own, if necessary.  It is highly recommended to include the search string (with correct annotations, brackets, etc) to better inform the reader of the specific search details.

2.  The type and/or credibility of each journal or outlet for studies included in the review (or exclusion criteria) needs to be provided and discussed.  Were these all peer-reviewed journal studies?  I am unable to tell in the methods section.

3.  Section 2.1.5 seems a little out of place.  I think it needs to go below after 2.1.6 to keep the systematic review methods together.

4.  Section 2.1.7 - I highly recommend summarizing the table's contents (table description) and referencing the table - not listing all table data within the manuscript paragraph.

5.  Section 2.1.8 - see #4 above.

6.  The meta-analysis discussion section is impressive.  Nice work!

7.  Please expand on the conclusion section significantly.  For instance - potential uses of the instrument/tool, opportunities for future research, etc.

8.  Please review the paper to ensure all acronyms are explained at their first appearance to assist with readability.  There are acronyms in the manuscript without descriptions (in parentheses) at all.

Thank you for considering these recommendations.

Round 2

Reviewer 2 Report

Minor suggestions

Line 50-53: The first sentence starting “Moreover, a safe…”is contradicted by the following sentence where you say that “applying potentially effective interventions to the children who will benefit most is critical to reducing the incidence of BPD and improving the prognosis.”  Consider writing.

Discussion:

2nd sentence: What does “it” refer to on line 339?  Consider re-writing for clarity.
